# Hunting for Discriminatory Proxies
# in Linear Regression Models

**Samuel Yeom**
Carnegie Mellon University
syeom@cs.cmu.edu

**Anupam Datta**
Carnegie Mellon University
danupam@cmu.edu

**Matt Fredrikson**
Carnegie Mellon University
mfredrik@cs.cmu.edu

## Abstract

A machine learning model may exhibit discrimination when used to make decisions involving people. One potential cause for such outcomes is that the model uses a statistical *proxy* for a protected demographic attribute. In this paper we formulate a definition of *proxy use* for the setting of linear regression and present algorithms for detecting proxies. Our definition follows recent work on proxies in classification models, and characterizes a model's constituent behavior that: *1)* correlates closely with a protected random variable, and *2)* is causally influential in the overall behavior of the model. We show that proxies in linear regression models can be efficiently identified by solving a second-order cone program, and further extend this result to account for situations where the use of a certain input variable is justified as a "business necessity". Finally, we present empirical results on two law enforcement datasets that exhibit varying degrees of racial disparity in prediction outcomes, demonstrating that proxies shed useful light on the causes of discriminatory behavior in models.

## 1 Introduction

The use of machine learning in domains like insurance [23], criminal justice [18], and child welfare [28] raises concerns about fairness, as decisions based on model predictions may discriminate on the basis of demographic attributes like race and gender. These concerns are driven by high-profile examples of models that appear to have discriminatory effect, ranging from gender bias in job advertisements [10] to racial bias in same-day delivery services [21] and predictive policing [3].

Meanwhile, laws and regulations in various jurisdictions prohibit certain practices that have discriminatory effect, regardless of whether the discrimination is intentional. For example, the U.S. has recognized the doctrine of *disparate impact* since 1971, when the Supreme Court held in *Griggs v. Duke Power Co.* [25] that the Duke Power Company had discriminated against its black employees by requiring a high-school diploma for promotion when the diploma had little to do with competence in the new job. These regulations pose a challenge for machine learning models, which may give discriminatory predictions as an unintentional side effect of misconfiguration or biased training data. Many competing definitions of disparate impact [3, 14] have been proposed in efforts to address this challenge, but it has been shown that some of these definitions are impossible to satisfy simultaneously [8]. Therefore, it is important to find a workable standard for detecting discriminatory behavior in models.

Much prior work [19, 30] has focused on the four-fifths rule [17] or variants thereof, which are relaxed versions of the *demographic parity* requirement that different demographic groups should receive identical outcomes on average. However, demographic parity does not necessarily make a model fair. For example, consider an attempt to "repair" a racially discriminatory predictive policing model by arbitrarily lowering the risk scores of some members of the disadvantaged race until demographic parity is reached. The resulting model is still unfair to individual members of

the disadvantaged race that did not have their scores adjusted. In fact, this is why the U.S. Supreme Court ruled that demographic parity is not a complete defense to claims of disparate impact [26]. In addition, simply enforcing demographic parity without regard for possible justifications for disparate impact may be prohibited on the grounds of intentional discrimination [27].

Recent work on *proxy use* [11] addresses these issues by considering the causal factors behind discriminatory behavior. A proxy for a protected attribute is defined as a portion of the model that is both causally influential [13] on the model's output and statistically associated with the protected variable. This means that, in the repair example above, the original discriminatory model is a proxy for a protected demographic attribute, indicating the presence of discriminatory behavior in the "repaired" model. However, prior treatment of proxy use has been limited to classification models, so regression models remain out of reach of these techniques.

In this paper, we define a notion of proxy use (Section 2) for linear regression models, and show how it can be used to inform considerations of fairness and discrimination. While the previous notion of proxy use is prohibitively expensive to apply at scale to real-world models [11], our definition admits a convex optimization procedure that leads to an efficient detection algorithm (Section 3). Because disparate impact is not always forbidden, we extend our definition to account for an *exempt* input variable whose use for a particular problem is justified. We show that slight modifications to our detection algorithm allow us to effectively "ignore" proxies based on the exempt variable (Section 4).

Finally, in Section 5 we evaluate our algorithm with two real-world predictive policing applications. We find that the algorithm, despite taking little time to run, accurately identifies parts of the model that are the most problematic in terms of disparate impact. Moreover, in one of the datasets, the strongest nonexempt proxy is significantly weaker than the strongest general proxy, suggesting that proxy use can sometimes be attributed to a single input variable. In other words, the proxies identified by our approach effectively explain the cause of discriminatory model predictions, informing the consideration of whether the disparate impact is justified.

Proofs of all theorems are given in the extended version of this paper [29].

## 1.1 Related Work

We refer the reader to [6] for a detailed discussion of discrimination in machine learning from a legal perspective. One legal development of note is the adoption of the four-fifths rule by the U.S. Equal Employment Opportunities Commission in 1978 [17]. The four-fifths "rule" is a guideline that compares the rates of favorable outcomes among different demographic groups, requiring that the ratio of these rates be no less than four-fifths. This guideline motivated the work of Feldman et al. [19], who guarantee that no classifier will violate the four-fifths rule by removing the association between the input variables and the protected attribute. Zafar et al. [30] use convex optimization to find linear models that are both accurate and fair, but their fairness definition, unlike ours, is derived from the four-fifths rule. We show in Section 2.5 that proxy use is a stronger notion of fairness than *demographic parity*, of which the four-fifths rule is a relaxation.

Other notions of fairness have been proposed as well. Dwork et al. [16] argue that demographic parity is insufficient as a fairness constraint, and instead define *individual fairness*, which requires that similar individuals have similar outcomes. While individual fairness is important, it is not well-suited for characterizing disparate impact, which inherently involves comparing different demographic groups to each other. Hardt et al. [20] propose a notion of group fairness called *equalized odds*. Notably, equalized odds does not require demographic parity, i.e., groups can have unequal outcomes as long as the response variable is also unequally distributed. For example, in the context of predictive policing, it would be acceptable to categorize members of a certain racial group as a higher risk on average, provided that they are in fact more likely to reoffend. This is consistent with the current legal standard, wherein disparate impact can be justified if there is an acceptable reason. However, some have observed that the response variable could be tainted by past discrimination [6, Section I.B.1], in which case equalized odds may end up perpetuating the discrimination.

Our treatment of exempt input variables is similar to that of resolving variables by Kilbertus et al. [22] in their work on causal analysis of proxy use and discrimination. A key difference is that they assume a causal model and only consider causal relationships between the protected attribute and the output of the model, whereas we view any association with the protected attribute as suspect. Our notion of proxy use extends that of Datta et al. [11, 12], who take into consideration both

*association* and *influence*. An alternative measure of proxy strength has been proposed by Adler et al. [1], who define a single real-valued metric called *indirect influence*. As we show in the rest of this paper, the two-metric-based approach of Datta et al. leads to an efficient proxy detection algorithm.

## 2  Proxy Use

In this section we present a definition of proxy use that is suited to linear regression models. We first review the original definition of Datta et al. [11] for classification models and then show how to modify this definition to get one that is applicable to the setting of linear regression.

### 2.1  Setting

We work in the standard machine learning setting, where a model is given several inputs that correspond to a data point. Throughout this paper, we will use $\mathcal{X} = (X_1, \ldots, X_n)$ to denote these inputs, where $X_1, \ldots, X_n$ are random variables. We consider a linear regression model $\hat{Y} = \beta_1 X_1 + \cdots + \beta_n X_n$, where $\beta_i$ represents the coefficient for the input variable $X_i$. We will abuse notation by using $\hat{Y}$ to represent either the model or its output.

In the case where each data point represents a person, care must be taken to avoid disparate impact on the basis of a protected demographic attribute, such as race or gender. We will denote such protected attribute by the random variable $Z$. In practice, $Z$ is usually binary (i.e., $Z \in \{0, 1\}$), but our results are general and apply to arbitrary numerical random variables.

### 2.2  Proxy Use in Prior Work

Datta et al. [11] define proxy use of a random variable $Z$ as the presence of an intermediate computation in a program that is both statistically *associated* with $Z$ and causally *influential* on the final output of the program. Instantiating this definition to a particular setting therefore entails specifying an appropriate notion of "intermediate computation", a statistical association measure, and a causal influence measure.

Datta et al. identify intermediate computations in terms of syntactic decompositions into *subprograms* $P$, $\hat{Y}'$ such that $\hat{Y}(\mathcal{X}) \equiv \hat{Y}'(\mathcal{X}, P(\mathcal{X}))$. Then the association between $P$ and $Z$ is given by an appropriate measure such as mutual information, and the influence of $P$ on $\hat{Y}$ is defined as shown in Equation 1, where $\mathcal{X}$ and $\mathcal{X}'$ are drawn independently from the population distribution.

$$\mathrm{Infl}_{\hat{Y}}(P) = \Pr_{\mathcal{X}, \mathcal{X}'}[\hat{Y}(\mathcal{X}) \neq \hat{Y}'(\mathcal{X}, P(\mathcal{X}'))], \tag{1}$$

Intuitively, influence is characterized by the likelihood that an independent change in the value of $P$ will cause a change in $\hat{Y}$. This makes sense for classification models because a change in the model's output corresponds to a change in the predicted class of a point, as reflected by the use of 0-1 loss in that setting. On the other hand, regression models have real-valued outputs, so the square loss is more appropriate for these models. Therefore, we are motivated to transform Equation 1, which is simply the expected 0-1 loss between $\hat{Y}(\mathcal{X})$ and $\hat{Y}'(\mathcal{X}, P(\mathcal{X}'))$, into Equation 2, which is the expected square loss between these two quantities.

$$\mathbb{E}_{\mathcal{X}, \mathcal{X}'}[(\hat{Y}(\mathcal{X}) - \hat{Y}'(\mathcal{X}, P(\mathcal{X}')))^2] \tag{2}$$

Before we can reason about the suitability of this measure, we must first define an appropriate notion of intermediate computation for linear models.

### 2.3  Linear components

The notion of subprogram used for discrete models [11] is not well-suited to linear regression. To see why, consider the model $\hat{Y} = \beta_1 X_1 + \beta_2 X_2 + \beta_3 X_3$. Suppose that this is computed using the grouping $(\beta_1 X_1 + \beta_2 X_2) + \beta_3 X_3$ and that the definition of subprogram honors this ordering. Then, $\beta_1 X_1 + \beta_2 X_2$ would be a subprogram, but $\beta_1 X_1 + \beta_3 X_3$ would not be even though $\hat{Y}$ could have been computed equivalently as $(\beta_1 X_1 + \beta_3 X_3) + \beta_2 X_2$. We might attempt to address this by allowing any subset of the terms used in the model to define a subprogram, thus capturing the commutativity and associativity of addition. However, this definition still excludes expressions such

as $\beta_1 X_1 + 0.5\beta_3 X_3$, which may be a stronger proxy than either $\beta_1 X_1$ or $\beta_1 X_1 + \beta_3 X_3$. To include such expressions, we present Definition 1 as the notion of subprogram that we use to define proxy use in the setting of linear regression.

**Definition 1** (Component). *Let* $\hat{Y} = \beta_1 X_1 + \cdots + \beta_n X_n$ *be a linear regression model. A random variable $P$ is a* component *of $\hat{Y}$ if and only if there exist* $\alpha_1, \ldots, \alpha_n \in [0, 1]$ *such that* $P = \alpha_1 \beta_1 X_1 + \cdots + \alpha_n \beta_n X_n$.

## 2.4 Linear association and influence

Having defined a component as the equivalent of a subprogram in a linear regression model, we now formalize the association and influence conditions given by Datta et al. [11].

**Association.** A linear model only uses linear relationships between variables, so our association measure only captures linear relationships. In particular, we use the Pearson correlation coefficient, and we square it so that a higher association measure always represents a stronger proxy.

**Definition 2** (Association). *The* association *of two nonconstant random variables $P$ and $Z$ is defined as* $\mathrm{Asc}(P, Z) = \frac{\mathrm{Cov}(P,Z)^2}{\mathrm{Var}(P)\mathrm{Var}(Z)}$.

Note that $\mathrm{Asc}(P, Z) \in [0, 1]$, with 0 representing no linear correlation and 1 representing a fully linear relationship.

**Influence.** To formalize influence, we continue from where we left off with Equation 2. Definition 1 gives us $\hat{Y}(\mathcal{X}) = \sum_{i=1}^{n} \beta_i X_i$ and $\hat{Y}'(\mathcal{X}, P(\mathcal{X}')) = \sum_{i=1}^{n}(1 - \alpha_i)\beta_i X_i + \alpha_i \beta_i X_i'$. Substituting these into Equation 2 gives

$$\mathbb{E}_{\mathcal{X},\mathcal{X}'}[(\hat{Y}(\mathcal{X}) - \hat{Y}'(\mathcal{X}, P(\mathcal{X}')))^2] = \mathbb{E}_{\mathcal{X},\mathcal{X}'}[(\sum_{i=1}^{n} \alpha_i \beta_i X_i - \alpha_i \beta_i X_i')^2] = \mathrm{Var}(P(\mathcal{X}) - P(\mathcal{X}')),$$

which is proportional to $\mathrm{Var}(P(\mathcal{X}))$ since $\mathcal{X}$ and $\mathcal{X}'$ are i.i.d. Definition 3 captures this reasoning, normalizing the variance so that $\mathrm{Infl}_{\hat{Y}}(P) = 1$ when $P = \hat{Y}$ (i.e., $\alpha_1 = \cdots = \alpha_n = 1$). In the extended version of this paper [29], we also show that variance is the unique influence measure (up to a constant factor) satisfying some natural axioms that we call nonnegativity, nonconstant positivity, and zero-covariance additivity.

**Definition 3** (Influence). *Let $P$ be a component of a linear regression model $\hat{Y}$. The* influence *of $P$ is defined as* $\mathrm{Infl}_{\hat{Y}}(P) = \frac{\mathrm{Var}(P)}{\mathrm{Var}(\hat{Y})}$.

When it is obvious from the context, the subscript $\hat{Y}$ may be omitted. Note that influence can exceed 1 because the inputs to a model can cancel each other out, leaving the final model less variable than some of its components.

Finally, the definition of proxy use for linear models is given in Definition 4.

**Definition 4** (($\epsilon, \delta$)-Proxy Use). *Let* $\epsilon, \delta \in (0, 1]$. *A model* $\hat{Y} = \beta_1 X_1 + \cdots + \beta_n X_n$ *has* ($\epsilon, \delta$)-proxy use *of $Z$ if there exists a component $P$ such that* $\mathrm{Asc}(P, Z) \geq \epsilon$ *and* $\mathrm{Infl}_{\hat{Y}}(P) \geq \delta$.

## 2.5 Connection to Demographic Parity

We now discuss the relationship between proxy use and demographic parity, and argue that proxy use is a stronger definition that provides more useful information than demographic parity. For binary classification models with two demographic groups, demographic parity is defined by the equation $\Pr[\hat{Y} = 1 | Z = 0] = \Pr[\hat{Y} = 1 | Z = 1]$, i.e., two demographic groups must have the same rates of favorable outcomes. We adapt this notion to regression models by replacing the constraint on the positive classification outcome with the expectation of the response, as shown in Definition 5.

**Definition 5** (Demographic Parity, Regression). *Let $\hat{Y}$ be a regression model, and let $Z$ be a binary random variable. $\hat{Y}$ satisfies* demographic parity *if* $\mathbb{E}[\hat{Y} | Z = 0] = \mathbb{E}[\hat{Y} | Z = 1]$.

Equation 3 shows that our association measure is related to demographic parity in regression models.

$$\mathrm{Asc}(\hat{Y}, Z) = \frac{\mathrm{Cov}(\hat{Y}, Z)^2}{\mathrm{Var}(\hat{Y})\mathrm{Var}(Z)} = (\mathbb{E}[\hat{Y} | Z = 0] - \mathbb{E}[\hat{Y} | Z = 1])^2 \cdot \frac{\mathrm{Var}(Z)}{\mathrm{Var}(\hat{Y})}, \tag{3}$$

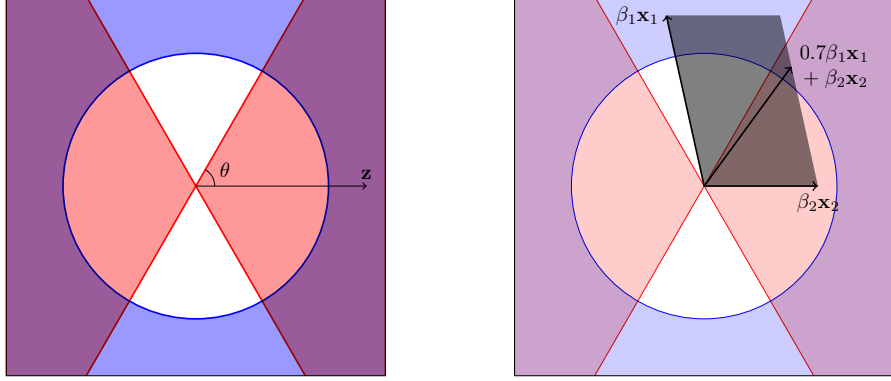

(a) $\mathbf{z}$ is a vector representation of the protected attribute $Z$, and components of the model can also be represented as vectors. If a component is inside the red double cone, it exceeds the association threshold $\epsilon$, where the angle $\theta$ is set such that $\epsilon = \cos^2 \theta$. The cone on the right side corresponds to positive correlation with $Z$, and the left cone negative correlation. Components in the blue shaded area exceed some influence threshold $\delta$. If any component exceeds both the association and the influence thresholds, it is a proxy and may be disallowed.

(b) $\mathbf{x}_1$ and $\mathbf{x}_2$ are vector representations of $X_1$ and $X_2$, which are inputs to the model $\hat{Y} = \beta_1 X_1 + \beta_2 X_2$. The gray shaded area indicates the space of all possible components of the model. $\beta_1 X_1$ is a component, but it is not a proxy because it does not have strong enough association with $Z$. Although $\beta_2 X_2$ is strongly associated with $Z$, it is not influential enough to be a proxy. On the other hand, $0.7\beta_1 X_1 + \beta_2 X_2$ is a component that exceeds both the association and the influence thresholds, so it is a proxy and may be disallowed.

Figure 1: Illustration of proxy use with the vector interpretation of random variables. In the above examples, all vectors lie in $\mathbb{R}^2$ for ease of depiction. In general, the vectors $\mathbf{z}, \mathbf{x}_1, \ldots, \mathbf{x}_n$ can span $\mathbb{R}^{n+1}$.

In particular, if $\hat{Y}$ does not satisfy demographic parity, then $\mathrm{Asc}(\hat{Y}, Z) > 0$, so $\hat{Y}$ is an $(\epsilon, 1)$-proxy for some $\epsilon > 0$. This means that our proxy use framework is broad enough to detect any violation of demographic parity. On the other hand, the "repair" example in Section 1 shows that demographic parity does not preclude the presence of proxies. Therefore, proxy use is a strictly stronger notion of fairness than demographic parity.

Moreover, instances of proxy use can inform the discussion about a model that exhibits demographic disparity. When a proxy is identified, it may explain the cause of the disparity and can help decide whether the behavior is justified based on the set of variables used by the proxy. We elaborate on this idea in Section 4, designating a certain input variable as always permissible to use.

## 3  Finding Proxy Use

In this section, we present our proxy detection algorithms, which take advantage of properties specific to linear regression to quickly identify components of interest. We prove that we can use an exact optimization problem (Problem 1) to either identify a proxy if one exists, or definitively conclude that there is no proxy. However, because this problem is not convex and in some cases may be intractable, we also present an approximate version of the problem (Problem 2) that sacrifices some precision. The approximate algorithm can still be used to conclude that a model does not have any proxies, but it may return false positives. In Section 5, we evaluate how these algorithms perform on real-world data.

Because the only operations that we perform on random variables are addition and scalar multiplication, we can safely treat the random variables as vectors in a vector space. In addition, covariance is an inner product in this vector space. As a result, it is helpful to think of random variables $Z, X_1, \ldots, X_n$ as vectors $\mathbf{z}, \mathbf{x}_1, \ldots, \mathbf{x}_n \in \mathbb{R}^{n+1}$, with covariance as dot product. Under this interpretation, influence is characterized by $\mathrm{Infl}_{\hat{Y}}(P) \propto \mathrm{Var}(P) = \mathrm{Cov}(P, P) = \mathbf{p} \cdot \mathbf{p} = \|\mathbf{p}\|^2$, where $\|\cdot\|$ denotes the $\ell^2$-norm, and association is shown in Equation 4, where $\theta$ is the angle

| **Problem 1** Exact optimization | **Problem 2** Approximate optimization |
|---|---|
| $\min \quad -\|A'\boldsymbol{\alpha}\|^2$ <br> $\text{s.t.} \quad 0 \preceq \boldsymbol{\alpha} \preceq 1 \text{ and } \|A'\boldsymbol{\alpha}\| \leq s \cdot \dfrac{\mathbf{z}^T A'\boldsymbol{\alpha}}{\sqrt{\epsilon}\|\mathbf{z}\|}$ | $\min \quad -\mathbf{c}^T\boldsymbol{\alpha}$ <br> $\text{s.t.} \quad 0 \preceq \boldsymbol{\alpha} \preceq 1 \text{ and } \|A'\boldsymbol{\alpha}\| \leq s \cdot \dfrac{\mathbf{z}^T A'\boldsymbol{\alpha}}{\sqrt{\epsilon}\|\mathbf{z}\|}$ |

Figure 2: Optimization problems used to find proxies in linear regression models. $A'$ is the $(n+1) \times n$ matrix $[\beta_1\mathbf{x}_1 \quad \ldots \quad \beta_n\mathbf{x}_n]$, and we optimize over $\boldsymbol{\alpha}$, which is an $n$-dimensional vector of the alpha-coefficients used in Definition 1. $\epsilon$ is the association threshold, and $\mathbf{c}$ is the $n$-dimensional vector that satisfies $c_i = \|\beta_i\mathbf{x}_i\|$.

between the two vectors $\mathbf{p}$ and $\mathbf{z}$.

$$\text{Asc}(P,Z) = \frac{\text{Cov}(P,Z)^2}{\text{Var}(P)\text{Var}(Z)} = \left(\frac{\mathbf{p} \cdot \mathbf{z}}{\|\mathbf{p}\|\|\mathbf{z}\|}\right)^2 = \cos^2\theta, \tag{4}$$

This abstraction is illustrated in more detail in Figure 1.

To find coordinates for the vectors, we consider the covariance matrix $[\text{Cov}(X_i, X_j)]_{i,j \in \{0,\ldots,n\}}$, where $Z = X_0$ for notational convenience. If we can write this covariance matrix as $A^T A$ for some $(n+1) \times (n+1)$ matrix $A$, then each entry in the covariance matrix is the dot product of two (not necessarily distinct) columns of $A$. In other words, the mapping from the random variables $Z, X_1, \ldots, X_n$ to the columns of $A$ preserves the inner product relationship. Now it remains to decompose the covariance matrix into the form $A^T A$. Since the covariance matrix is guaranteed to be positive semidefinite, two of the possible decompositions are the Cholesky decomposition and the matrix square root.

Our proxy detection algorithms use as subroutines the optimization problems that are formally stated in Figure 2. We first motivate the exact optimization problem (Problem 1) and show how the solutions to these problems can be used to determine whether the model contains a proxy. Then, we present the approximate optimization problem (Problem 2), which sacrifices exactness for efficient solvability.

Let $A'$ be the $(n+1) \times n$ matrix $[\beta_1\mathbf{x}_1 \quad \ldots \quad \beta_n\mathbf{x}_n]$. The constraint $0 \preceq \boldsymbol{\alpha} \preceq 1$ restricts the solutions to be inside the space of all components, represented by the gray shaded area in Figure 1b. Moreover, when $s \in \{-1, 1\}$, the constraint $\|A'\boldsymbol{\alpha}\| \leq s \cdot (\mathbf{z}^T A'\boldsymbol{\alpha})/(\sqrt{\epsilon}\|\mathbf{z}\|)$ describes one of the red cones in Figure 1a, which together represent the association constraint. Subject to these constraints, we maximize the influence, which is proportional to $\|A'\boldsymbol{\alpha}\|^2$. Theorem 1 shows that this technique is sufficient to determine whether a model contains a proxy.

**Theorem 1.** *Let $P$ denote the component defined by the alpha-coefficients $\boldsymbol{\alpha}$. The linear regression model $\hat{Y} = \beta_1 X_1 + \cdots + \beta_n X_n$ contains a proxy if and only if there exists a solution to Problem 1 with $s \in \{-1, 1\}$ such that $\text{Infl}_{\hat{Y}}(P) \geq \delta$.*

In essence, Theorem 1 guarantees the correctness of the following proxy detection algorithm: Run Problem 1 with $s = 1$ and $s = -1$, and compute the association and influence of the resulting solutions. The model contains a proxy if and only if any of the solutions passes both the association and the influence thresholds.

It is worth mentioning that Problem 1 tests for strong positive correlation with $Z$ when $s = 1$ and for strong negative correlation when $s = -1$. This optimization problem resembles a second-order cone program (SOCP) [7, Section 4.4.2], which can be solved efficiently. However, the objective function is concave, so the standard techniques for solving SOCPs do not work on this problem. To get around this issue, we can instead solve Problem 2, which has a linear objective function whose coefficients $c_i = \|\beta_i\mathbf{x}_i\|$ were chosen so that the inequality $\|A'\boldsymbol{\alpha}\| \leq \mathbf{c}^T\boldsymbol{\alpha}$ always holds. This inequality allows us to prove Theorem 2, which mirrors the claim of Theorem 1 but only in one direction.

**Theorem 2.** *If the linear regression model $\hat{Y} = \beta_1 X_1 + \cdots + \beta_n X_n$ contains a proxy, then there exists a solution to Problem 2 with $s \in \{-1, 1\}$ such that $\mathbf{c}^T\boldsymbol{\alpha} \geq (\delta \text{Var}(\hat{Y}))^{0.5}$.*

Theorem 2 suggests a quick algorithm to verify that a model does not have any proxies. We solve the SOCP described by Problem 2, once with $s = 1$ and once with $s = -1$. If neither solution

satisfies $\mathbf{c}^T \boldsymbol{\alpha} \geq (\delta \operatorname{Var}(\hat{Y}))^{0.5}$, by the contrapositive of Theorem 2, we can be sure that the model does not contain any proxies.

However, the converse does not hold, i.e., we cannot be sure that the model has a proxy even if a solution to Problem 2 satisfies $\mathbf{c}^T \boldsymbol{\alpha} \geq (\delta \operatorname{Var}(\hat{Y}))^{0.5}$. This is because $\mathbf{c}^T \boldsymbol{\alpha}$ overapproximates $\|A'\boldsymbol{\alpha}\|$ by using the triangle inequality. As a result, it is possible for the influence to be below the threshold even if the value of $\mathbf{c}^T \boldsymbol{\alpha}$ is above the threshold. While there is in general no upper bound on the overapproximation factor of the triangle inequality, the experiments in Section 5 show that this factor is not too large in practice. In addition, Problem 1 often works well enough in practice despite not being a convex optimization problem.

## 4  Exempt Use of a Variable

So far, we have shown how to find a proxy in a linear regression model, but we have not discussed which proxies should be allowed and which should not. As mentioned in Section 1, disparate impact is legally permitted if there is sufficient justification. For example, in the context of predictive policing, it may be acceptable to consider the number of prior convictions even if one racial group tends to have a higher number of convictions than another. We formalize this idea by assuming that the use of one particular input variable, which we call the *exempt variable*, is explicitly permitted. This assumption may be appropriate if, for example, the exempt variable is directly and causally related to the response variable $Y$. Throughout this section, we will use $X_1$ to denote the exempt variable.

First, we formally define which proxies are *exempt*, i.e., permitted because the proxy use is attributable to $X_1$. Clearly, if the model ignores every input except $X_1$, all proxies in the model should be exempt. Conversely, if the coefficient $\beta_1$ of $X_1$ is zero, no proxies should be exempt. We capture this intuition by ignoring $X_1$ and checking whether the resulting component is a proxy. More formally, if $P = \alpha_1\beta_1 X_1 + \cdots + \alpha_n\beta_n X_n$ is a component, we investigate $P \setminus X_1$, which we write as shorthand for the component $\alpha_2\beta_2 X_2 + \cdots + \alpha_n\beta_n X_n$. If $P$ is a proxy but $P \setminus X_1$ is not, then $P$ is exempted because the proxy use can be attributed to the exempt variable $X_1$.

However, one possible issue with this attribution is that the other input variables can interact with $X_1$ to create a proxy stronger than $X_1$. For example, suppose that $\operatorname{Asc}(X_2, Z) = 0$ and $P = X_1 + X_2 = Z$. Then, even though $P \setminus X_1 = X_2$ is not a proxy, it makes $P$ more strongly associated with $Z$ than $X_1$ is, so it is not clear that $P$ should be exempt on account of the fact that we are permitted to use $X_1$. Therefore, our definition of proxy exemption in Definition 6 adds the requirement that $P$ should not be too much more associated with $Z$ than $X_1$ is.

**Definition 6** (Proxy Exemption)**.** *Let $P$ be a proxy component of a linear regression model, and let $X_1$ be the exempt variable. $P$ is an* exempt *proxy if $P \setminus X_1$ is not a proxy and $\operatorname{Asc}(P, Z) < \operatorname{Asc}(X_1, Z) + \epsilon'$, where $\epsilon'$ is the association tolerance parameter.*

We can incorporate the exemption policy into our search algorithm with small changes to the optimization problem. By Definition 6, a proxy $P$ is nonexempt if either $P \setminus X_1$ is a proxy or $\operatorname{Asc}(P, Z) \geq \operatorname{Asc}(X_1, Z) + \epsilon'$. For each of these two conditions, we modify the optimization problems from Section 3 to find proxies that also satisfy the condition. If either of these modifications return a positive result, then we have found a nonexempt proxy.

We start with the second condition, for which it is easy to see that it suffices to change the association threshold in Problem 2 from $\epsilon$ to $\max(\epsilon, \operatorname{Asc}(X_1, Z) + \epsilon')$. For the first condition, we use the result from Theorem 3 and simply add the constraint that $\alpha_1 = 0$. If we add this constraint to Problem 2, the resulting problem is still an SOCP and can therefore be solved efficiently.

**Theorem 3.** *A linear regression model contains a proxy $P$ such that $P \setminus X_1$ is also a proxy if and only if the model contains a proxy such that $\alpha_1 = 0$.*

## 5  Experimental Results

In this section, we evaluate the performance of our algorithms on real-world predictive policing datasets. We ran our proxy detection algorithms on observational data from Chicago's Strategic Subject List (SSL) model [9] and the Communities and Crimes (C&C) dataset [15]. The creator of the SSL model claims that the model avoids variables that could lead to discrimination [4], and if

| Association threshold $\epsilon$ | 0.01 | 0.02 | 0.03 | 0.04 | 0.05 | 0.06 | 0.07 |
|---|---|---|---|---|---|---|---|
| Actual infl. (Prob. 1) | 0.8816 | 0.2263 | 0.1090 | 0.0427* | 0.0065* | 0.0028 | 0.0000 |
| Approx. infl. (Prob. 2) | 1.6933 | 0.6683 | 0.3820 | 0.1432 | 0.0270 | 0.0085 | 0.0000 |
| Actual infl. (Prob. 2) | 0.8476 | 0.1874 | 0.0987 | 0.0420 | 0.0080 | 0.0027 | 0.0000 |

Table 1: Influence of the components obtained by solving the exact (Problem 1) and approximate (Problem 2) optimization problems for the SSL model using $Z = \mathrm{race}$ and $s = 1$. No component had strong enough association when $s = -1$ instead. Asterisks indicate that the exact optimization problem terminated early due to a singular KKT matrix. The approximate optimization problem did not have this issue, and the overapproximation that it makes of the components' influence is shown in the second row.

this is the case then we would expect to see only weak proxies if any. On the other hand, the C&C dataset contains many variables that are correlated with race, so we would expect to find strong proxies in a model trained with this dataset.

To test these hypotheses, we implemented Problems 1 and 2 with the `cvxopt` package [2] in Python. The experimental results confirm our hypotheses and show that our algorithm runs very quickly ($< 1$ second). Moreover, our algorithms pinpoint components of the model that are the most problematic in terms of disparate impact, and we find that the exemption policy discussed in Section 4 removes the appropriate proxies from the SSL model.

For each dataset, we briefly describe the dataset and present the experimental results, demonstrating how the identified proxies can provide evidence of discriminatory behavior in models. Then, we explain the implications of these results on the false positive and false negative rates in practice, and we discuss how a practitioner can decide which values of $\epsilon$ and $\delta$ to use.

**Strategic Subject List.** The SSL [9] is a model that the Chicago Police Department uses to assess an individual's risk of being involved in a shooting incident, either as a victim or a perpetrator. The SSL dataset consists of 398,684 rows, each of which corresponds to a person. Each row includes the SSL model's eight input variables (including age, number of previous arrests for violent offenses, and whether the person is a member of a gang), the SSL score given by the model, and the person's race and gender.

We searched for proxies for race (binary black/white) and gender (binary male/female), filtering out rows with other race or gender. After also filtering out rows with missing data, we were left with 290,085 rows. Because we did not have direct access to the SSL model, we trained a linear regression model to predict the SSL score of a person given the same set of variables that the SSL model uses. Our model explains approximately 80% of the variance in the SSL scores, so we believe that it is a reasonable approximation of the true model for the purposes of this evaluation.

The strengths of the proxies for race are given in Table 1. The estimated influence was computed as $(\mathbf{c}^T\boldsymbol{\alpha})^2/\mathrm{Var}(\hat{Y})$, which is the result of solving for $\delta$ in the inequality given in Theorem 2. We found that this estimate is generally about 3–4$\times$ larger than the actual influence. Although the proxies for race were somewhat stronger than those for gender, neither type had significant influence ($\delta > 0.05$) beyond small $\epsilon$ levels (~0.03–0.04). This is consistent with our hypothesis about the lack of discriminatory behavior in this model.

We also tested the effect of exempting the indicator variable for gang membership in the input. Gang membership is more associated with both demographic variables than any other in among the inputs, and is a plausible cause of involvement in violent crimes [5], making it a prime candidate for exemption. As contrasted with the components described in Table 1, every nonexempt component under this policy has an association with race less than 0.033. This means that the strongest nonexempt proxy is significantly weaker than the strongest general proxy, suggesting that much of the proxy use present in the model can be attributed to the gang membership variable.

**Communities and Crimes.** C&C [24] is a dataset in the UCI machine learning repository [15] that combines socioeconomic data from the 1990 US census with the 1995 FBI Uniform Crime Reporting data. It consists of 1,994 rows, each of which corresponds to a community (e.g.,

municipality) in the U.S., and 122 potential input variables. After we removed the variables that directly measure race and the ones with missing data, we were left with 90 input variables.

We simulated a hypothetical naive attempt at predictive policing by using this dataset to train a linear regression model that predicts the per capita rate of violent crimes in a community. We defined the protected attribute $Z$ as the difference between the percentages of people in the community who are black and white, respectively. We observed a strong association in the dataset between the rate of violent crime and $Z$ ($\mathrm{Asc}(Y, Z) = 0.48$), and the model amplifies this bias even more ($\mathrm{Asc}(\hat{Y}, Z) = 0.65$).

As expected, we found very strong proxies for race in the model trained with the C&C dataset. For example, one proxy consisting of 58 of the 90 input variables achieves an influence of 0.34 when $\epsilon = 0.85$. Notably, the input variable most strongly associated with race has an association of only 0.73, showing that *in practice multiple variables combine to result in a stronger proxy than any of the individual variables.* In addition, the model contains a proxy whose association is 0.40 and influence is 14.5. In other words, the variance of the proxy is *14.5 times greater than that of the model*; this arises because other associated variables cancel most of this variance in the full model. As a result, exempting any one variable does not result in a significant difference since associated variables still yield proxies that are nearly as strong. Moreover, a cursory analysis suggested that the variables used in these proxies are not justifiable correlates of race, so an exemption policy may not suffice to "explain away" the discriminatory behavior of the model.

**False Positives and False Negatives.** Theorem 1 shows that our exact proxy detection algorithm detects a proxy if and only if the model in fact contains a proxy. In other words, if Problem 1 returns optimal solutions, we can use the solutions to conclusively determine whether there exists a proxy, and there will be no false positives or false negatives. However, our experiments show that sometimes Problem 1 terminates early due to a singular KKT matrix, and in this case one can turn to the approximate proxy detection algorithm.

Although Problem 2 sometimes returns solutions that are not in fact proxies, we can easily ascertain whether any given solution is a proxy by simply computing its association and influence. However, even if the solution returned by Problem 2 turn out to not be proxies, the model could still contain a different proxy. Using Table 1 as reference, we see that this happens in the SSL model if, for example, $\epsilon = 0.02$ and $\delta$ is between 0.1874 and 0.2263. Therefore, one can consider the approximate algorithm as giving a finding of either "potential proxy use" or "no proxy use". Theorem 2 shows that a finding of "no proxy use" does indeed guarantee that the model is free of proxies. In other words, the approximate algorithm has no false negatives. However, the algorithm overapproximates influence, so the algorithm can give a finding of "potential proxy use" when there are no proxies, resulting in a false positive. This happens when $\delta$ is between the maximum feasible influence (first row in Table 1) and the maximum feasible overapproximation of influence (second row in Table 1).

**Reasonable Values of $\epsilon$ and $\delta$.** Although the appropriate values of $\epsilon$ and $\delta$ depend on the application, we remind the reader that association is the *square* of the Pearson correlation coefficient. This means that an association of 0.05 corresponds to a Pearson correlation coefficient of ~0.22, which represents not an insignificant amount of correlation. Likewise, influence is proportional to variance, which increases quadratically with scalar coefficients. Therefore, we recommend against setting $\epsilon$ and $\delta$ to a value much higher than 0.05. To get an idea of which values of $\delta$ are suitable for a particular application, the practitioner can compare the proposed value of $\delta$ against the influence of the individual input variables $\beta_i X_i$.

## 6    Conclusion and Future Work

In this paper, we have formalized the notion of proxy discrimination in linear regression models and presented an efficient proxy detection algorithm. We account for the case where the use of one variable is justified, and extending this result to multiple exempt variables is valuable future work that would enable better handling of models like C&C that take many closely related input variables. Developing learning rules that account for proxy use, leading to models without proxies above specified thresholds, is also an intriguing direction with direct potential for impact on practical scenarios.

**Acknowledgment**

The authors would like to thank the anonymous reviewers at NeurIPS 2018 for their thoughtful feedback. This material is based upon work supported by the National Science Foundation under Grant No. CNS-1704845.

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
