[Reviews · NeurIPS 2018]

Reviewer 1



This paper proposes a definition of proxy use in linear models and provides an algorithm for detecting such proxy use. Inspired by Datta et al. (2016, 2017), they define proxy use in terms of association and influence, where proxy discrimination is the existence of some linear combination of features that is highly associated with the protected attribute and has some causal influence on the outcome. The authors show that discovering such a linear combination with high association and influence can be relaxed to a SOCP, at the cost of losing one direction of tightness. In particular, they show that if a regression model contains a discriminatory proxy as they define it, then the SOCP that they provide will find it. Finally, they provide experimental results, showing that very strong proxies exist in both datasets they consider. In the SSL dataset, most of this is due to the "gang membership" variable; however, in the C&C dataset, proxies remain strong even when exempting individual variables, suggesting that in that dataset, many variables participate as proxies, making it difficult to prevent their use. The techniques provided in this paper seem to provide a fairly interesting way to analyze datasets and understand the relations between the features they contain. While the scope of this work is limited to linear relations and may not be applicable to the actual models used in practice, it still could be useful in discovering which variables might be serving as proxies.

Reviewer 2



Summary This paper describes a framework for detecting proxy variables in a linear regression framework. It poses the problem as two optimization problems and presents (with proofs only in supplemental material) theorems that relate the solutions to the two optimization problems to cases of proxy existence in a problem. The paper also describes incorporation of an exempt variable, a proxy that is deemed acceptable for use for one reason or another. The paper leverages a prior work that defines a proxy in a classification framework as a variable that is associated with a sensitive attriute and causally infulential on the decision of the system. The paper describes how to reformulate this definition for the case of linear regression. The proposed method is tested on two datasets and finds expected results. Strengths: The text of the paper is generally understandable. Proxy detection can be an important problem in checking for discriminatory behavior of an algorthmic system. Weaknesses: The presentation of material sometimes reads abruptly, some reorganization or trimming of tangential points might make the main points of the paper easier to ascertain for the reader. For example, section three begins by introducing an abstraction, stating that it will be used throughout while describbing the proxy detection algorithm, but then key details of the algorithm are left out. It skips directly to solving an optimization problem, the origin of which is never described, but would be more informative than the geometric interpretation/abstraction, as it reads currently. Perhaps a description of the optimization problems would be faciiated by the abstraction, but as is it reads like a tangent. Problem 1 and problem 2 are referred to prior to definiton and never defined in the body of the paper; only in table headings and figure captions. The table (figure 2) where they are defined is also never explicitly referred to in the body of the paper. Testing a linear regression set of techniques on data that is described as having largely categoricla varialbes doesn't seem like a sensible choice. Either a better description of the data, justifying that linear regression is the right technique for the the data or better data set selection is necessary to validate that the experimental results are meaningful and that the current technique (as opposed to the prior work that motivates this work) is necessary.

Reviewer 3



This paper describes an optimization based approach to identify variables that result in "discrimination by proxy" in linear regression models. The authors present a definition of proxy variables, proposes an optimization problem that can be solved to identify proxies, and present a surrogate problem that can be solved efficiently. They extend the results of this approach to settings where a "proxy" variable is exempt, and illustrate how an "exempt" variable can be strengthened. The paper includes experiments on two real-world datasets, which show that the defensible" proxies are generates. PROS - Well-motivated problem / novel approach. - Well-founded definition of a proxy for linear regression models. CONS - Reliability of detection is not considered - Experiments on real-world data do not validate the method (as there is no ground truth) - No guidance in how to set a key parameter for proxy detection (i.e., the association threshold \epsilon)? Major Comments: ------------- This paper considers a well-motivated problem and proposes an interesting solution. However, I am concerned about several shortcomings in the current manuscript: 1. There is no discussion about how to control false positive / false negatives in proxy detection? If I were to implement their approach, and use it to detect proxies, then the first thing that I would like to have is some degree of certainty regarding how often I am detecting / not detecting proxies. I suspect that this is far more relevant and should be prioritized. 2. There is no empirical validation of the proposed method. The authors include experiments on real-world dataset, but these do not validate the method because we do not know the ground truth in these problems (i.e., the results are not falsifiable and just "make sense") Both 1 and 2 affect the significance of the work. Both issues could be addressed by adding experiments on a toy problem where we know the ground truth. I'd advise designing a simple simulated regression problem where they know the ground truth. The proposed method should be able to correctly detect a proxy when it exists and avoid detecting a proxy when it does not exist. This would be a good way to discuss the potential to misreport proxy use. Post Rebuttal ------------- I would like to thank the authors for the work that they put into their paper and the rebuttal. Many of the main concerns in my original review were addressed within the rebuttal. As a result, I am raising my score to a 7. I expect that the authors will incorporate the FPR/FNR discussion into the text (as it is important for practitioners). The experiments should also be included as they provide some more convincing evidence. In what follows, I provide some additional comments that can be quickly addressed to improve the paper. There are not intended as criticisms of the work, but simply intended to strengthen the paper in the long run: 1. Provide guidance in how to choose a key hyperarameter for proxy detection (i.e., the association threshold \epsilon)? Some key questions to answer include: - Should practitioners pick a single value of epsilon? Or run the analysis for different values of epsilon? In the latter case, will there be conflicting results? How should be choose? - Can epsilon be bounded after looking at a particular dataset? I think that bounds could be derived from Problem 1. If epsilon is too large, then \beta = 0 is the only feasible solution. - What should a practitioner expect as they increase the value of \epsilon? 2. The paper is missing a discussion on how your work could be used. Say that we have identified a proxy, now what? 3. The related work is too broad. The entire discussion on fair classification methods (e.g., Dwork/Zafar et al) could be replaced with a statement that says: "Our approach aims to explain why a linear regression model may discriminate by proxy. Other works aim to produce classifiers that do not discriminate as defined by a specific definition of fairness [cite fair classification work]." Instead you may want to cite other papers that are focused on proxy detection. 4. The writing could be improved in later parts of the paper. Some tips. - Text describing notation could be far shorter. For example, l. 103 - 106 could be replaced with "We consider a linear regression model \hat{Y} = \sum_j \alpha_j X_j" where "\alpha_j" represents the coefficient for variable j. - There is some colloquial language that can be corrected. For example: - l. 181 catch <- detect - l. 278 stays away <- does not include - l. 152 Note that we always have <- Note that - Datta et al. [11, 12] <- cite both in the related works, but only refer to one in the technical section (so we only have to check one paper) - If variables use a norm, then define the type of norm